# Household Cases Suggest That Cats Belonging to Owners with COVID-19 Have a Limited Role in Virus Transmission

**DOI:** 10.3390/v13040673

**Published:** 2021-04-14

**Authors:** Pierre Bessière, Maxime Fusade-Boyer, Mathilda Walch, Laetitia Lèbre, Jessie Brun, Guillaume Croville, Séverine Boullier, Marie-Christine Cadiergues, Jean-Luc Guérin

**Affiliations:** 1IHAP, Université de Toulouse, INRAE, ENVT, 31300 Toulouse, France; maxime.fusade-boyer@envt.fr (M.F.-B.); mathilda.walch@envt.fr (M.W.); laetitia.lebre@envt.fr (L.L.); guillaume.croville@envt.fr (G.C.); jean-luc.guerin@envt.fr (J.-L.G.); 2Small Animal Clinic, Université de Toulouse, ENVT, 31300 Toulouse, France; jessie.brun@envt.fr (J.B.); marie-christine.cadiergues@envt.fr (M.-C.C.); 3InTheRes, Université de Toulouse, INRAE, ENVT, 31300 Toulouse, France; severine.boullier@envt.fr; 4Infinity, Université de Toulouse, INSERM, CNRS, UT3, ENVT, 31300 Toulouse, France

**Keywords:** COVID-19, cats, reverse-zoonosis

## Abstract

Severe acute respiratory syndrome coronavirus-2 (SARS-CoV-2) is responsible for COVID-19 and spread rapidly following its emergence in Wuhan in 2019. Although cats are, among other domestic animals, susceptible to SARS-CoV-2 infection, little is known about their epidemiological role in the dynamics of a household infection. In this study, we monitored five cats for viral shedding daily. Each cat was confined with its COVID-19 positive owners in separate households. Low loads of viral nucleic acid were found in two cats, but only one developed anti-SARS-CoV-2 antibodies, which suggests that cats have a limited role in COVID-19 epidemiology.

## 1. Introduction

Since its emergence in late 2019, SARS-CoV-2, the etiological agent of COVID-19, has infected millions of people throughout the world. The virus spread from an animal reservoir, probably of bats, and might have involved an intermediate host. This highlights the importance of animals and the One Health perspective in the control of a zoonotic and emerging disease [1]. In the past few months, spillover events have resulted in several countries reporting a viral transmission from humans to animals. These events mostly involved cats [2,3,4,5,6], but also dogs [6,7] and minks [8]. Since other domestic animals, such as ferrets [9] and hamsters [10], are also susceptible to SARS-CoV-2 infection, reverse zoonosis events involving these animals will likely occur. In most case reports involving cats, animals are sampled only once and few studies perform daily monitoring of viral shedding. Human-to-cat transmission might be underestimated and there is a strong need to better understand the role of cats in COVID-19 epidemiology [11]. In this study, we monitored viral shedding in five different cats living in five different households with their quarantined and COVID-19-positive owners.

## 2. Materials and Methods

### 2.1. Owners and Cats

All of the owners developed COVID-19-like symptoms (fever, cough, headache, fatigue, and a loss of taste or smell) and subsequently tested positive for SARS-CoV-2 infection by means of RT-qPCR performed on nasopharyngeal swabs in a medical facility in accordance with the French governmental guidelines [12]. They all belonged to clusters involving friends and relatives who were never in contact with the cats. Four owners were veterinary students, while a cat from case 5 was owned by a veterinarian’s relative. All of the cats lived strictly indoors. For one week during their quarantine, the owners performed daily oropharyngeal and rectal swabs on their cats and the cats’ food bowls (environmental swabs) following veterinary instructions (EC approval SSA_2020_0010). These instructions were given by phone and through a document, available in Appendix A. Swabs were collected daily and stored dry at 4 °C in the meantime. The delay between the onset of the owners’ symptoms and the first cat swabbing ranged from four to eight days. Blood samples were taken from all of the cats at least six weeks after their owners’ recovery to assess the cats’ serological status. For more details about the owners, see Appendix A.

### 2.2. RT-qPCR from Feline and Environmental Swabs

Swabs were briefly vortexed in 500 µL PBS and viral RNA was extracted from 140 µL in accordance with the manufacturer’s instructions (QIamp viral RNA; Qiagen, Toronto, ON, Canada). RNA extractions were performed within the 48 h following sampling (swabs were kept at 4 °C in the meantime). RT-qPCR was performed on 96-well plates with a final volume of 20 µL in a Light Cycler system (Roche, Penzberg, Germany). The mixes were prepared in accordance with the manufacturer’s instructions (QuantiNova Probe; Qiagen, Toronto, ON, Canada) with 2 µL of RNA and primers that targeted the E-gene. To obtain human-derived viral RNA and perform phylogenetic analysis, the owner from case 3 self-performed a nasal swab which was treated as a feline one. To check whether the material in the oropharyngeal and rectal swabs originated from felines, RT-qPCR targeting the feline 40S ribosomal protein S7 (RPS7) gene was also performed. SARS-CoV-2-positive swabs were further tested for human ribosomal protein L30 (RPL30) mRNA expression, to rule out human contamination. RPS7 and RPL30 RT-qPCR were performed on 96-well plates in a final volume of 10 µL in a Light Cycler System (Roche, Penzberg, Germany). Mixes were prepared according to the manufacturer’s instructions (QuantiFast; Qiagen, Toronto, ON, Canada). The primers’ sequences are listed in Appendix A.

### 2.3. Virus Isolation

Vero E6 cells were plated in a 12-well plate and cultured in Dulbecco’s modified Eagle medium (DMEM) and complemented with 10% of heat-inactivated fetal bovine serum and 1% of penicillin-streptomycin at 37 °C and with 5% of CO_2_. Cells at 90% confluence were incubated with 100 µL of each RT-qPCR positive viral sample in DMEM complemented the 2% of heat-inactivated fetal bovine serum with 1% of penicillin-streptomycin, 2 µg/mL of ciprofloxacin, 10 µg/mL of BM-cyclin 1, and 1 µg/mL of amphotericin B (Merck, Darmstadt, Germany) for 72 h. Infectivity was assessed by screening cells for cytopathic effects and, at the end of the 72-h waiting period, the presence of SARS-CoV-2 nucleic acids was confirmed by RT-qPCR. Experiments were carried out in a biosafety level 3 facility at the National Veterinary School of Toulouse.

### 2.4. ELISA

The serological status of the cats was assessed using a commercial SARS-CoV-2 N double antigen enzyme-linked immunosorbent assay (ELISA) kit in accordance with the manufacturer’s instructions (ID Screen; ID-Vet, Montpellier, France).

### 2.5. Serum Neutralization Assay

Vero-E6 cells were plated in 96-plates and cultured DMEM complemented with 10% of heat-inactivated fetal bovine serum and 1% of penicillin-streptomycin at 37 °C and with 5% of CO_2_. The samples and controls were heat-inactivated at 56 °C for 30 min, serially diluted in DMEM starting at 1:10, mixed with 100 TCID_50_ of SARS-CoV-2 (previously amplified and titrated on Vero-E6 cells), incubated for 2 h at 37 °C, and transferred to a culture plate. Following a 1-h incubation at 37 °C, the virus-serum dilutions were removed, and the cells were washed with PBS. Cells were incubated in a growth medium (DMEM complemented with 2% of heat-inactivated fetal bovine serum and 1% of penicillin-streptomycin) for 72 h at 37 °C and with 5% of CO_2_ and were screened for cytopathic effects. Serum from a SARS-CoV-2 immunized mouse (kindly provided by JC. Guéry) was used as a positive control. PBS was used as a negative control. Experiments were carried out in a biosafety level 3 facility at the National Veterinary School of Toulouse.

### 2.6. Whole-Genome Sequencing

A SARS-CoV-2 whole-genome amplification was performed according to the ARTIC amplicon sequencing protocol for MinION for nCoV-2019 [13]. Briefly, reverse transcription of viral RNA was performed using an SSIV Reverse Transcriptase (Thermo Fisher Scientific, Inc., Mississauga, ON, Canada) and random hexamers followed by a PCR with the Q5 Hot Start DNA Polymerase (New England Biolabs, Ipswich, MA, USA) and the hCoV-2019/nCoV-2019 Version 3 Amplicon Set. Samples were submitted to this protocol and the amplification was visually checked by running 5 µL aliquots on 1% agarose gel. DNA libraries were prepared using an SQK-LSK109 Ligation sequencing kit supplied by ONT (Oxford Nanopore Technologies, Oxford, UK) associated with an EXP-NBD104 (ONT) native barcoding kit for the multiplexing of 3 samples. In total, 30 fmol DNA libraries were loaded on a FLO-FLG001 flongle and were run on a MinION Mk1C device (ONT) for 10 h. Fast base-calling was performed in real-time with Guppy (v3.5) embedded in the MK1C software (v19.12.12) with the ‘Trim Barcode’ option on (ONT). Fastq files were mapped on the MW420003.1 reference genome using bwa-sw (v0.7.15) and consensus genomes were produced with bcftools mpileup (v1.6). Consensus sequences were then checked using the Geneious (v.2019.03) software (Biomatters, Ltd., Auckland, New Zealand). The owner- and feline-derived viral sequences were submitted to GenBank (accession numbers MW505982.1 and MW513511.1 respectively).

### 2.7. Phylogenetic Analysis 

Whole-genome sequences of the 50 strains most closely related to hCoV-19/cat/France/LHOOQ/2020 and hCoV-19/human/FRA/66JQ-O/2020 (the strains isolated from case 3 and its owner, respectively) were downloaded from NCBI using a BLASTn search. All cat-isolated SARS-CoV-2 sequences also were loaded, as well as one strain representative of each clade according to GISAID classification. All of the sequences were aligned using the MAFFT online server and tested for recombination events using the Recombination Detection Program 4 (RDP4) [14]. When several of the collected sequences in the same host and place were similar, only one was kept for the following steps. Phylogenetic analysis was performed using a generalized time-reversible (GTR) model with discrete gamma distribution and four categories implemented in MegaX (MEGA X: Molecular Evolutionary Genetics Analysis across computing platforms; Kumar, Stecher, Li, Knyaz, and Tamura 2018) performing a 1000-bootstrap resampling analysis—a process visualized with Figtree (v.1.4.2).

## 3. Results

### 3.1. Case Description

Five neutered European cats (two females and three males) belonging to owners with COVID-19 were enrolled in this study. The animals ranged in age from 1 to 10 years old. None of the cats developed any symptoms. Age, breed, gender, and samples are recapitulated in Table 1.

### 3.2. Two Cats Out of Five Shed SARS-CoV-2

Cases 3 and 5 tested positive for SARS-CoV-2 RNA. For case 3, low quantities of viral nucleic acids were detected in a rectal swab (Ct 37.0) and an oropharyngeal swab (Ct 35.6) three and four days, respectively, after the owner’s diagnosis. Viral nucleic acids were also detected in the cat’s feeding bowl on days 5 and 6 after the owner’s diagnosis (Ct 33.6 and 34.2 respectively). For case 5, viral nucleic acid was detected once on an oropharyngeal swab (Ct 36.7) and once on an environmental swab (Ct 37.5) (Table 1). All cat and environmental swabs were tested for feline 40S ribosomal protein S7 (RPS7) mRNA expression to check whether swabbing was performed appropriately: feline RPS7 mRNA expression was detected in all feline swabs and most environmental swabs (including SARS-CoV-2-positive ones). SARS-CoV-2-positive samples were further tested for human ribosomal protein L30 (RPL30) mRNA expression, which was not detected, suggesting that the presence of viral RNA was not due to human contamination. Virus isolation from positive swabs on Vero E6 cells were unsuccessful. All other swabs were negative at the other time points.

### 3.3. One Cat Out of Five Developed Anti-SARS-CoV-2 Antibodies

Antibodies were found in case 3 only and the other samples were negative. Serum from case 3 yielded positive ELISA titers (OD value: 1.088; % positivity: 121%), which was confirmed by the detection of neutralizing antibodies (neutralization titer > 1:1280). These results suggest that SARS-CoV-2 was effectively transmitted in only this case, despite the proximity of the other owners with their cats.

### 3.4. Phylogenetic Analysis

A partial viral genome (accession number MW513511.1) was obtained from the positive oropharyngeal swab of case 3, while a whole genome (accession number MW505982.1) was obtained from the owner’s nasopharyngeal swab. Phylogenetic analysis performed on the whole genome showed that both viruses in our study had clustered with viruses from Switzerland and belonged to the clade GH according to GISAID nomenclature. This clade is characterized by the nucleotide substitutions C241T, C3037T, A23403G, G25563T, the amino acid substitution Q57H in the NS3 protein, and the substitution D614G in the spike protein, which is known to enhance infectivity and viral fitness [15]. Phylogenetic analysis also showed that both sequences from case 3 were very closely related, sharing more than 99.99% genetic identity (Figure 1). Both sequences differed only by the synonymous G26031A mutation and an amino acid substitution F843L in the ORF1ab protein that was not described in the literature. Nevertheless, since only a partial genome was obtained from the cat sample, two regions (from amino acid (aa) 262 to aa 318, and from aa 448 to aa 520) were not successfully sequenced and other differences may have been missed.

## 4. Discussion

Cats belonging to COVID-19-positive owners were monitored for viral shedding and clinical signs. None of the cats developed symptoms. SARS-CoV-2 nucleic acids were found by RT-qPCR in two cats, for two consecutive days in cat 3, and only once in cat 5, and both had high Cts values. They were also found on the dinner bowls of both cats, suggesting that the virus was shed in the house, although it could have emanated from the owners as well as from the cats. We detected RPS7 mRNA using RT-qPCR in all of the oropharyngeal and rectal swabs, demonstrating that the owners managed to swab their animals correctly. Furthermore, human RPL30 mRNA was not detected in any SARS-CoV-2-positive feline or environmental swabs, thus decreasing the probability that viral RNA presence was due to human contamination. Case 3 had the only cat that tested positive for anti-SARS-CoV-2 antibodies using both ELISA and serum virus neutralization assays. The other cats tested negative by means of RT-qPCR for SARS-CoV-2 and, according to serological tests made a few weeks later, the cats also tested negative for antibodies. This suggests that the cats were never infected, rather than that the shedding window was missed when swabbing the animals. Since viral shedding was weak and transient, cats may not be efficient at shedding SARS-CoV-2.

In experimental studies, cats inoculated intranasally or intratracheally with SARS-CoV-2 shed viruses up to 10 days post-infection [16,17] and were able to infect non-inoculated cats through direct or indirect contact [18]. However, the inoculum dose and the infection route may not reflect the natural circumstances in which a cat could be naturally infected. In a more recent study, three cats living together in one household with COVID-19-positive owners were monitored for viral shedding and the SARS-CoV-2 RNA was sequentially detected in all three cats—whether this was due to a cat-to-cat transmission remains unknown. Interestingly, viral RNA was detected up to 11 days after the onset of the owners’ symptoms [19].

In our study, the cats lived in small to moderate-sized households and the owners did not take particular hygiene measures while interacting with their pets. Despite these elements, in only two cats was viral RNA detected. Cat 5 did not develop anti-SARS-CoV-2 antibodies and a viral genome was found only once in an oropharyngeal swab and once in an environmental swab. It remains unknown whether the swabs were contaminated with viral RNA from the owner. In addition, human cases with transient SARS-CoV-2 shedding and an absence of antibody responses have been reported [20,21] and the same phenomenon could occur in cats. Notably, Temman and colleagues failed to detect anti-SARS-CoV-2 antibodies in nine cats belonging to COVID-19 positive students [22].

The infected cats did not display any clinical signs, which is consistent with previous experimental studies [16,17,18]. However, subclinical manifestations cannot be excluded and it is worth noting there have been some reports of symptomatic cats in households with COVID-19 patients [2,3,4,19]. Age and comorbidities are likely involved in these differences and further studies are required to better understand SARS-CoV-2 pathogenesis in cats.

In addition to the epidemiological context, phylogenetic analysis supports the human-to-cat transmission hypothesis since the strains detected in the owners and the cats shared a high genetic identity. The feline viral genome differed only by one amino-acid and one nucleotide substitution. A sequence comparison with the other available feline-derived SARS-CoV-2 genomes did not reveal any shared cat-specific mutation. None of the available viral sequences from cats harbored this substitution, which suggests a random event rather than an adaptation to cats despite the low number of available sequences.

Our study has a number of limitations. The sample size was limited due to the constraining nature of the study’s protocol, which renders it difficult to generalize our findings to a larger cat population. Due to the strict lockdown measures at the time of this study, the owners performed the swabbing unsupervised. They received detailed instructions, but we cannot exclude the possibility that the samples were not taken thoroughly enough, despite the RPS7 mRNA presence, or that they were contaminated by the owners’ hands or fomites, despite the human RPL30 mRNA absence. In addition, there is a lack of evidence regarding the sensitivity of oropharyngeal swabbing compared to nasal or nasopharyngeal swabbing in cats.

In conclusion, the SARS-CoV-2 genome was detected by RT-qPCR in two cats out of five, which suggests that human-to-cat transmission is not an infrequent event. Given the variability of clinical manifestations, COVID-19 in cats is likely under-diagnosed. However, viral shedding was weak and transient, which hints that, even when infected, cats probably play a limited role in COVID-19 epidemiology.

## Figures and Tables

**Figure 1 viruses-13-00673-f001:**
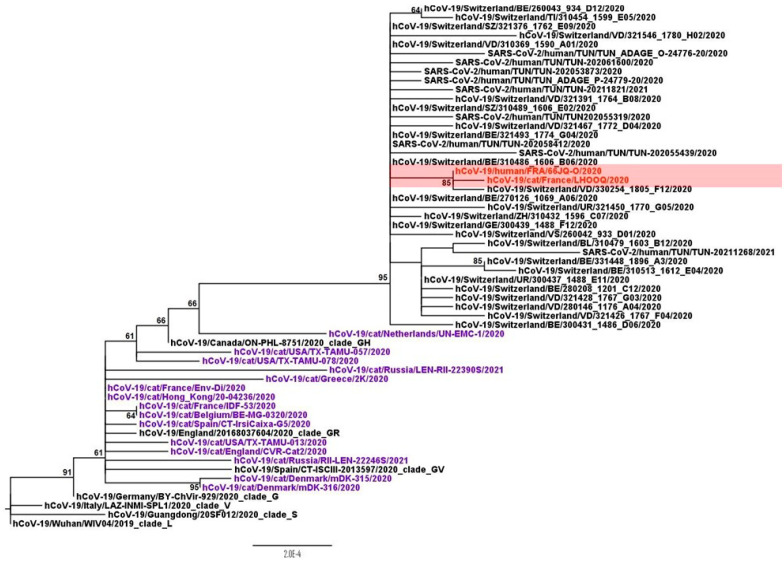
Whole-genome maximum likelihood phylogenetic tree of SARS-CoV-2. The viruses in Table 1, sequence from case 3, and the owner-derived sequence are both labelled in red. Other feline-derived viral sequences are labeled in purple. Bootstrap supports >60% are indicated next to the nodes; the scale bar indicates the numbers of nucleotide substitutions per site. The strain hCoV-19/Wuhan/WIV04/2019 was used as the outgroup.

**Table 1 viruses-13-00673-t001:** Summary of the cases, including breed, age, gender, the time between the onset of owner’s symptoms and cat’s first swab, number, and RT-qPCR Ct values of positive swabs (Day 1 corresponding to the first day the cat was enrolled in the study) and serological status determined by ELISA and serum neutralization assays.

Case	Breed	Age (Years)	Gender	Days Between Owners’ Symptoms onset and Cat Swabbing	Positive Swabs			ELISA/ SN (Days ^1^)
					OP	R	Env	
1	DSH	1	Female	4	None	None	None	Neg/Neg (54)
2	DSH	10	Male	6	None	None	None	Neg/Neg (62)
3	DSH	3	Female	5	Day 2: 35.6	Day 1: 35.7	Day 1: 33.6 Day 2: 34.2	Pos/Pos (47)
4	DSH	1	Male	8	None	None	None	Neg/Neg (51)
5	DSH	6	Male	7	Day 3: 36.7	None	Day 6: 37.5	Neg/Neg (174)

DSH: domestic shorthair cat; OP: oropharyngeal; R: rectal; Env: environment; Neg: negative; Pos: positive; SN: serum neutralization assay. ^1^: time between the onset of owners’ symptoms and the ELISA test.

## Data Availability

The data presented in this study are available on request from the corresponding author. The data are not publicly available due to the interests of retaining patient confidentiality.

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
