# Peer review of "Household Cases Suggest That Cats Belonging to Owners with COVID-19 Have a Limited Role in Virus Transmission"

_viruses, 2021, doi:10.3390/v13040673_

Round 1

Reviewer 1 Report

It is a very study interesting because it proposes the infectious follow-up of 5 cats in households where the owners were infected with SARS-CoV-2. Only few studies have been done with a daily monitoring of the cats. The results are well described and demonstrate the limited impact of infected cats in the viral SARS-CoV-2 propagation.

I only have minor corrections, to improve the quality of the manuscript:

1/ Could the authors give more details about the households ? Number of owners living with the cats? presence of other companion animals ?

2/ Could the authors precise the value obtained for the serological analyse of the positive cat ? (O.D value, % of positivity ?)

3/ Lines 165-166, please qualify your sentence as the study in reference 15 does not prove the cat-to cat transmission, it is a possiblity but there is no demonstration in this study.

Reviewer 2 Report

Since reports emerged of human-to-domestic cat transmission of SARS-CoV-2, the question has arisen about the frequency and likelihood of such transmission events. This study addresses this important question by monitoring five cats from 5 separate COVID-19 households for evidence of infection, using RT-qPCR to detect viral shedding and ELISA to detect anti-nucleocapsid antibodies.  Two cats tested positive for viral RNA but only one of them was seropositive. The authors concluded that cats appear to have a limited role in COVID-19 epidemiology.  

Ideally, to support this conclusion, it would be necessary to monitor a larger sample size, but nevertheless this study is important and could be used to calculate the required sample number for future analyses with greater statistical power. The methodology was sound, although the possibility cannot be excluded that the samples tested by RT-qPCR might have been contaminated by the owners who were required to collect the oropharyngeal swabs themselves during their lockdown. Nevertheless, the finding that virus shedding was weak and only transient is important.

Reviewer 3 Report

viruses-1150936

Although studies evaluating the possible role of pets in SARS CoV 2 transmission are needed, and the topic in interesting, the present study seems not rigorous enough to support conclusions rised from the authors.

Introduction

Line 29: actually, besides those reported by the authors, in literature are present several recent studies (some of them based on hundreds of samples)  focused on the possible role of dogs and cats in the epidemiology of COVID. Maybe some may be added

  • Patterson, E. I., et al. (2020). Evidence of exposure to SARS-CoV-2 in cats and dogs from households in Italy. Nature Communications, 11, 6231. doi: 10.1038/s41467-020-20097-0.
  • Hobbs, E. C. &, Reid, T. J. (2020). Animals and SARS-CoV-2: Species susceptibility and viral transmission in experimental and natural conditions, and the potential implications for community transmission. Transboundary and Emerging Diseases, doi: 10.1111/tbed.13885. Online ahead of print.
  • Kiros, M., et al. (2020). COVID-19 pandemic: current knowledge about the role of pets and other animals in disease transmission. Virology Journal, 17(1), 1-8. doi:10.1186/s12985-020-01416-9
  • Leroy, E. M., Gouilh, M. A. & Brugère‑Picoux, J. (2020). The risk of SARS-CoV-2 transmission to pets and other wild and domestic animals strongly mandates a one-health strategy to control the COVID-19 pandemic. One Health, 10:100133. doi:10.1016/j.onehlt.2020.100133
  • Pagani, et al. (2021). Human-to-Cat SARS-CoV-2 Transmission: Case Report and Full-Genome Sequencing from an Infected Pet and Its Owner in Northern Italy. Pathogens, 10(2), 252. doi:10.3390/pathogens10020252.

Line 32: It isn’t clear what “in separate households means”. Separate by who or what? Please, explain.

Materials and method

Line 34: please list symptoms or add a reference from literature (the expression “COVID-like” sounds ambiguous). Please define ‘subsequently’: how many days elapsed between symptoms and swabs?

Line 36: please add a reference about the method used to perform SARS-CoV-2 swabs diagnostics in people in France (also national guidelines or similar are acceptable)

Line 37: were owners veterinarians? It is very hard that a non-vet is able to perform oropharingeal and rectal swabs to conscious cats…could you please report (at least as supplementary material) a summary of instructions given to owners? How the authors could exclude that a contamination from the positive and symptomatic owners could have generated positive swabs in their cats by spreading the virus?

Line 41: is the RTqPCR method referred to all the samples or only the feline ones? Please, clarify. If the method is reported in literature, this should be added in the reference list. Was the laboratory a

What happened to swabs between the collection by the owner and the analyses? Were they stored by the owners? Were they frozen? For how many days? An RNA later solution or other preservatives were used? Please, add some details.

Is there any evidence that the ELISA is able to recognize feline antibodies or is it designed for people? Were neutralizing antibodies checked for accuracy? Are false positive reactions to other feline coronaviruses excluded?

Results

Lines 96-97: actually this information should be included in the M&M section not in the results.

Lines 100-102 “All oropharyngeal and rectal swabs were tested for cat 40S ribosomal protein S7 (RPS7)

mRNA expression as an internal control to check whether swabbing was performed appropriately”.

I don’t understand how the presence of feline material in cats is considered as internal control….It is quite obvious that some animal genetic material may be included in swabbing…actually the absence of human 40S ribosomal protein… should have been checked in order to exclude contamination from positive owners.

Were the feeding bowls never been touched by the positive owners? Were someone else besides positive owners feeding the cats?

Line 103: a Ct 37.0 is really low…and 99,9%  identity is really high…

All of these data make a contamination very likely, especially considering the authors do not report sufficient data to exclude it.

Discussion

Lines 146-147: these info should be included in the M&M.

Many conclusions are overinflated, considering that analyses performed on over 1000 pets owned by COVID patients in previous articles never detected viral RNA in swabs from cats. It is quite a wonderful coincidence that here 2/5 cats were positive and also spread the virus into the environment….they authors should be careful in their conclusions which are not supported by strong evidence of animal positivity instead of contamination from infected  owners…

Round 2

Reviewer 3 Report

Authors answered timely, correctly and thoroughly to all reviewer’s  comments. The addition of supplementary tests (SN, human housekeeping genes detection etc.) gives more robusteness to the study.

The manuscript is now eligible for publication. Few minor points:

Line 41: “while cat from case 5 was a veterinarian relative.” I would replace with “while cat from case 5 was owned by a veterinarian relative.”

Line 128: I would replace “Cases 3 and 5 tested positive for SARS-CoV-2 infection” with Cats 3 and 5 tested positive for SARS-CoV-2 RNA

Table 1: I would add here also results from SN for completeness of data reported

Line 182: I would replace “confirming that virus was shed in the house” with “suggesting that virus was shed in the house”

Line 206: I would replace “only two cats became productively infected” with “in only two cats viral RNA was detected”

Author Response

Dear reviewer,

We thank you for your positive comments on our revised manuscript. We added all your corrections on lines 41, 128 182, and 206 (which are now lines 41, 128, 184, and 208). We also added SN to table 1.

Sincerely,

Pierre Bessiere (in the name of all authors)